# Simulation of the Impact of Firebrands on the Process of the Wood Layer Ignition

Oleg Matvienko [1,2], Denis Kasymov [2,*], Egor Loboda [2], Anastasia Lutsenko [2] and Olga Daneyko [1,2]

[1] Department of Automobile Roads, Tomsk State University of Architecture and Building, 634003 Tomsk, Russia; matvolegv@mail.ru (O.M.); olya_dan@mail.ru (O.D.)
[2] Department of Physical and Computational Mechanics, Tomsk State University, 634050 Tomsk, Russia; loboda@mail.tsu.ru (E.L.); anastas_mex_mat434@mail.ru (A.L.)
[*] Correspondence: kdp@mail.tsu.ru

**Abstract:** In this study, a theoretical formulation of the ignition and combustion of the wood layer by burning and smoldering firebrands has been considered. The effect of the firebrands' length, distances between firebrands and their geometrical parameters on the heat exchange with the wood layer and the ignition process were analyzed. With a decrease in firebrand size, ignition of wood is possible with a decrease in the distance between the firebrands. With an increase in firebrand size at the same distance between them, the ignition regime becomes possible albeit with a longer delay time Δt. With a decrease in the distance between the firebrands, the ignition of wood is possible with an increase in Δt. As a result of mathematical modeling of the process, the following processes are noted: the heat stored in firebrands of small sizes is insufficient to initiate the ignition process; the temperature in the wood layer, due to conductive heat exchange, slightly increases at first, before beginning to decrease as a result of heat exchange with the surrounding air and the wood layer; intensive heat exchange with the environment of small size firebrands leads to the end of firebrand smoldering and its cooling; and, if the firebrand size reaches a critical value, then the pyrolysis process begins in the area adjacent to it.

**Keywords:** firebrands; wood; combustion; mathematical modeling; heat transfer; pyrolysis





## 1. Introduction

Forest ecosystems are subject to numerous destructive impacts, such as wildland fires, mass reproduction of insect pests, deforestation, and other natural and anthropogenic factors. Wildland fires have a dominant negative impact on the state and dynamics of forest ecosystems [1,2], causing significant material and environmental damage.

Burning and smoldering firebrands of natural origin are the particles that have been generated as a result of heating and deformation of flammable natural materials, such as shrubs, trees (or any other combustible natural material), or building materials [3]. One of the effects observed in large-scale wildfires is burning and smoldering firebrands formed in the fire front. They are capable of covering a distance of several kilometers and initiating a new combustion center [4].

Recently, studies have been actively conducted that aim to systematically investigate the formation of firebrands, including both full-scale studies of the ignition of building structures [5–8] and modeling of the interaction of firebrands with structures based on wood [9–16]. The mechanism of ignition and combustion of the single firebrand model of various shapes, as well as the interaction of such a point source of thermal exposure with forest fuels, has been studied detailly in past studies [17–22].

For some time, attempts have been made to quantify the critical conditions for ignition caused by the accumulation of burning and smoldering firebrands. The thermal characteristics of a pile of firebrands on an inert plate were studied in a number of works [16,23,24]. These experiments showed that heating from a pile of firebrands is affected not only by the

speed of the surrounding wind, but also by the mass and bulk density of the accumulated firebrands. It is, thus, of interest to study the process of firebrand accumulation near various types of building and structural materials to assess its tendency to ignite. In particular, a series of experiments was carried out in a past study [25] with 9 firebrands (3 by 3 square) applied to birch plywood 6.35 mm thick. The mass of the firebrands was 13–15 g. The distance between firebrands varied from 10 to 30 mm. The firebrands were ignited and placed on plywood. Various wind speeds were taken: 0, 0.05, and 0.75 m/s. The results show that firebrands with a spacing of 20 mm or more were able to burn through only the plywood surface until completely burning out. When the distance between the firebrands is less than 20 mm, plywood ignites and continues to burn even after the firebrands have completely burned out.

It was concluded that a single firebrand cannot cause plywood to ignite under the chosen parameters of the experiment. Firebrands located at a distance of 30 mm or more from each other cannot cause a sustained fire either. When the distance between the firebrands decreased, the plywood ignited and the fire continued to grow even when the firebrands stopped burning. This effect was also observed by Kwon and Liao [26]. In these experiments, firebrands were simulated using nine wooden cubes, with 19 mm on each side. The results showed that the flame height and the sample mass loss rate had non-monotonic dependencies based on the gap spacing. For smoldering combustion, compared to a single burning sample, the smoldering temperature and duration significantly increased due to the thermal interactions between adjacent burning samples [27].

Another study [28] presented an experimental and numerical study of how the thermal interactions of the fuel bed affect the subsequent tendency to ignite wood in contact with two electric heaters acting as idealized firebrands. The authors tried to find out the critical conditions (distance between firebrands, size, etc.) at which two idealized firebrands together can ignite solid wood fuel. Two sizes of heaters were used in the study: width 7.5 and 15 mm; length 25 and 35 mm, respectively.

Experimental studies show that for two heaters of both sizes, reducing the distance between them leads to the appearance of igniting ability. It is shown that the proximity of two heaters not only leads to an increase in the range of conditions under which ignition occurs, but also reduces the process of fuel ignition (dry white pine wood was used as fuel). In addition, the authors experimentally confirm that when several firebrands are placed in close proximity or collected in compact piles, an increase in the number of firebrands creates the opportunity for more heat to be released, which is subsequently transferred to the wood. Moreover, the accumulated firebrands can isolate each other, causing most of their energy to be transferred to the fuel rather than the environment. Thus, the accumulation of combustible mixture is considered to act as a more favorable source of ignition, co-igniting the target fuel and resulting in faster or more probable ignition compared to a similar situation with only one combustible mixture.

Apart from laboratory experiments, large-scale experimental studies were carried out to study the ability to ignite structural fuel by piles of burning and smoldering fire-brands [29–32]. Some studies have used versions of the firebrand generator [33–35] to spray firebrands onto the fuel and observe ignition during their accumulation. Under certain conditions, individual or small piles of firebrands can ignite receptive fuels [36]; in other cases, they can accumulate and ignite larger fuels, such as wood covering [29]. The ability of single firebrands [20,37–39] or idealized firebrands [40–42] to ignite fuel was also studied in laboratory studies. The igniting possibilities of piles of firebrands to ignite a structural fuel have been investigated in laboratory experiments [16,42–46] that characterized the heat transfer from a pile of firebrands to the fuel and the subsequent ignition process. Heat exchange studies showed peak heat fluxes up to 70 kW/m$^2$ transferred from smoldering firebrands [16,23,24,47,48]. Some studies have shown that piles of firebrands not only form a large heated area, but also cause a higher heat flux to the fuel [16,23,24]. Numerical or analytical studies of fuel heating and/or ignition by firebrands have been carried out, but they have mainly focused on single firebrands [21,22,38,49]; one study also considered

small compact piles [39]. Several numerical studies assessed firebrands involvement in the co-ignition of various fuel types [50–52].

Numerous experiments [27,53,54] have been carried out on full-size structural elements, such as decks, fences, and building corners, subjected to firebrands' shower using the Fire Dragon. The authors found that these constructions ignited more easily from a pile of firebrands than a single firebrand. It was shown in past studies [23,48,55,56] that the condition for ignition by a pile and a single firebrand differ.

It is necessary to study the ignition process for firebrands of specific sizes; a single hit on the wood layer will not lead to ignition. Indeed, if a single firebrand is able to initiate the process of ignition and combustion, the hit of a combination of such firebrands on the wood layer will also cause its ignition. Thus, in mathematical modeling it was assumed that the ignition of the wood layer by a single firebrand does not occur. However, its ignition becomes possible when two firebrands hit the surface. The fulfillment of this assumption was ensured by the choice of the appropriate firebrand sizes.

The present study focuses on studying wood layer ignition as a result of the accumulation of model firebrands simulating pine twigs. The novelty concludes in the investigation of the ignition process by firebrands falling on the surface non-simultaneously.

## 2. Materials and Methods

*Mathematical Model*

The ignition and combustion of the wood layer by burning and smoldering firebrands was considered. It was assumed that the combustible components, which made up the volatile products of pyrolysis, could be modeled by one effective combustible gas with the reactive properties of carbon monoxide. The rectangular coordinate system was used for a mathematical description: the OX and OY axes were directed horizontally, while the OZ axis was directed vertically upwards (Figure 1).

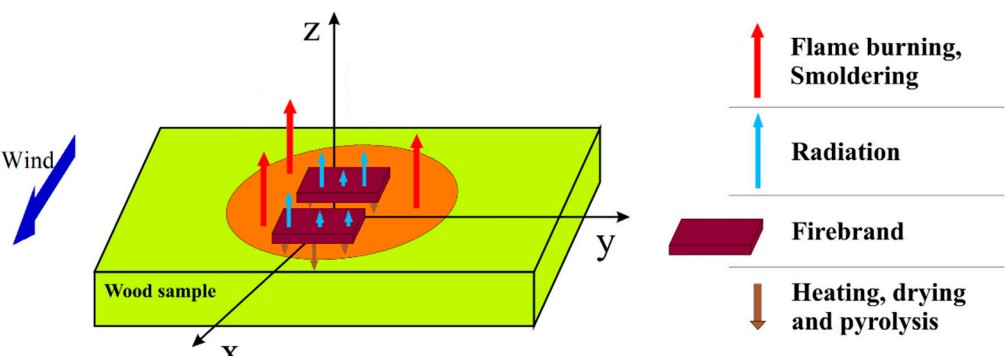

**Figure 1.** Process diagram.

The balance of thermal energy in a smoldering firebrand and a layer of wood is described by the thermal conductivity equation. This equation has the form for a smoldering firebrand [57]:

$$C_{fb}\frac{\partial \rho_{fb}T_{fb}}{\partial t} = \frac{\partial}{\partial x}\left[\lambda_f \frac{\partial T_{fb}}{\partial x}\right] + \frac{\partial}{\partial y}\left[\lambda_f \frac{\partial T_{fb}}{\partial y}\right] + \frac{\partial}{\partial z}\left[\lambda_f \frac{\partial T_{fb}}{\partial z}\right] + Q_{td}\Phi_{td} + Q_{pyr}\Phi_{pyr} + \varepsilon^* \frac{\left(U - \sigma T_{fb}^4\right)}{\delta}. \tag{1}$$

The thermal conductivity equation for the wood layer can be written as:

$$C_w\frac{\partial \rho_w T_w}{\partial t} = \frac{\partial}{\partial x}\left[\lambda_w \frac{\partial T_w}{\partial x}\right] + \frac{\partial}{\partial y}\left[\lambda_w \frac{\partial T_w}{\partial y}\right] + \frac{\partial}{\partial z}\left[\lambda_w \frac{\partial T_w}{\partial z}\right] + Q_{td}\Phi_{td} - Q_{dr}\Phi_{dr} + Q_{pyr}\Phi_{pyr} + \varepsilon^* \frac{\left(U - \sigma T_w^4\right)}{\delta}, \tag{2}$$

where $C, \rho, \lambda, \varepsilon^*$ are the heat capacity, density, thermal conductivity, and emissivity of the condensed phase. Index $fb$ is related to the firebrand, $w$—to the wood layer.

Radiation heat transfer is determined by the $P_1$ approximation. The radiation intensity is determined from the equation solution [57,58]:

$$\frac{\partial^2 U}{\partial x^2} + \frac{\partial^2 U}{\partial y^2} + \frac{\partial^2 U}{\partial z^2} = 3\beta\chi\left(U - \sigma T_g^4\right), \tag{3}$$

where $\beta$ is the integral radiation absorption coefficient, $\chi$ is the integral radiation attenuation coefficient, and $\sigma$ is the black body radiation constant.

Evaporation of moisture associated with the wood layer can be described by the following dependence [59]:

$$\Phi_{dr} = k_{dr}\rho_f M_{hum}T^{-2.25}exp\left(-\frac{T_{dr}}{T_f}\right), \tag{4}$$

where $k_{dr} = 6\cdot10^5 K^{2.25}/s^{-1}$, $T_{dr} = 6\cdot10^4 K$ is a pre-exponential factor and an evaporation activation temperature. The heat of evaporation reaction was assumed to be $k_{dr} = 3\cdot10^6$ J/kg. $M_{hum}$ is the relative wood humidity.

The pyrolysis mass rate of the condensed phase (firebrand and layer of wood) can be calculated using the dependence [57,59]:

$$\Phi_{pyr} = k_{pyr}\rho_f M_1 exp\left(-\frac{T_{pyr}}{T_f}\right) \tag{5}$$

where thermokinetic parameters have the following values: $k_{pyr} = 3.63\cdot10^4 s^{-1}$, $T_{pyr} = 9.4\cdot10^3 K$, $Q_{pyr} = 10^7$ J/kg. $M_1$ is the mass fraction of dry organic material, $f = fb$ for igniting firebrands, and $f = w$ for the wood layer.

The combustion rate of condensed pyrolysis products are described by the following dependence [57,59]:

$$\Phi_{td} = k_{td}\rho_f M_{O_2} M_{pyr} exp\left(-\frac{T_{td}}{T_f}\right) \tag{6}$$

where $k_{td} = 10^6 s^{-1}$, $T_{td} = 10^4 K$—pre-exponential index and temperature of the heterophase combustion reaction activation. The heat of combustion reaction of condensed pyrolysis products is taken equal to $Q_{dr} = 1.2\cdot10^7$ J/kg. $M_{pyr}$ is the mass fraction of condensed pyrolysis products.

The mass change in the wood layer, water change in a liquid-drop state, and change in the condensed pyrolysis products are described by the balance equations:

$$\frac{\partial\rho_f M_1}{\partial t} = -\Phi_{pyr}, \quad \frac{\partial\rho_f M_2}{\partial t} = -\Phi_{dr}, \quad \frac{\partial\rho_f M_3}{\partial t} = \Phi_{pyr} - \Phi_{td} \tag{7}$$

The thermal conductivity equation of the gas phase can be written as [60–63]:

$$C_{p,g}\left(\frac{\partial\rho_g T_g}{\partial t} + \frac{\partial\rho_g v_x T_g}{\partial x} + \frac{\partial\rho_g v_y T_g}{\partial y} + \frac{\partial\rho_g v_z T_g}{\partial z}\right) = \frac{\partial}{\partial x}\left[\lambda_g\frac{\partial T_g}{\partial x}\right] + \frac{\partial}{\partial y}\left[\lambda_g\frac{\partial T_g}{\partial y}\right] + \frac{\partial}{\partial z}\left[\lambda_g\frac{\partial T_g}{\partial z}\right] + Q_g\Phi_g + \dot{h} + \beta\left(U - \sigma T_g^4\right) \tag{8}$$

The enthalpy change in the gas phase due to the influx of gaseous pyrolysis products and water vapor into it, which are formed during the drying of wood, can be calculated as follows:

$$\dot{h} = \dot{h}_{pyr} + \dot{h}_{dr}, \quad h_{pyr} = C_{p,CO}\left(T_f - T_g\right)\left(\Phi_{pyr} - \Phi_{td}\right), \quad \dot{h}_{dr} = C_{p,H_2O}\left(T_f - T_g\right)\Phi_{dr}. \tag{9}$$

The chemical reaction rate in the gas phase is described by the dependence [64,65]:

$$\Phi_g = k_g\rho_g^2 M_{O_2} M_{CO}exp\left(-\frac{E_g}{R_g T_g}\right) \tag{10}$$

The chemical reaction parameters were taken in accordance with the data of two past studies [66,67]: $k_g = 2.2 \cdot 10^8 \cdot m^3 \cdot kg^{-1} \cdot s^{-1}$, $E_g = 104 \ kJ \cdot mol^{-1}$.

The mass balance equations for the components $O_2, CO, CO_2, N_2, H_2O$ were used [68–70] to describe the processes of diffusion, mixing, chemical reaction, and combustion in the gas phase, in addition to the energy equation:

$$\frac{\partial \rho_g M_{CO}}{\partial t} + \frac{\partial \rho_g v_x M_{CO}}{\partial x} + \frac{\partial \rho_g v_y M_{CO}}{\partial y} + \frac{\partial \rho_g v_z M_{CO}}{\partial z} = \frac{\partial}{\partial x}\left[\rho_g D \frac{\partial M_{CO}}{\partial x}\right] + \frac{\partial}{\partial y}\left[\rho_g D \frac{\partial M_{CO}}{\partial y}\right] + \frac{\partial}{\partial z}\left[\rho_g D \frac{\partial M_{CO}}{\partial z}\right] - 2\frac{W_{CO}}{W_{O_2}}\Phi_g + \Phi_{pyr}, \quad (11)$$

$$\frac{\partial \rho_g M_{O_2}}{\partial t} + \frac{\partial \rho_g v_x M_{O_2}}{\partial x} + \frac{\partial \rho_g v_y M_{O_2}}{\partial y} + \frac{\partial \rho_g v_z M_{O_2}}{\partial z} = \frac{\partial}{\partial x}\left[\rho_g D \frac{\partial M_{O_2}}{\partial x}\right] + \frac{\partial}{\partial y}\left[\rho_g D \frac{\partial M_{O_2}}{\partial y}\right] + \frac{\partial}{\partial z}\left[\rho_g D \frac{\partial M_{O_2}}{\partial z}\right] - \Phi_g, \quad (12)$$

$$\frac{\partial \rho_g M_{CO_2}}{\partial t} + \frac{\partial \rho_g v_x M_{CO_2}}{\partial x} + \frac{\partial \rho_g v_y M_{CO_2}}{\partial y} + \frac{\partial \rho_g v_z M_{CO_2}}{\partial z} = \frac{\partial}{\partial x}\left[\rho_g D \frac{\partial M_{CO_2}}{\partial x}\right] + \frac{\partial}{\partial y}\left[\rho_g D \frac{\partial M_{CO_2}}{\partial y}\right] + \frac{\partial}{\partial z}\left[\rho_g D \frac{\partial M_{CO_2}}{\partial z}\right] + 2\frac{W_{CO_2}}{W_{O_2}}\Phi_g, \quad (13)$$

$$\frac{\partial \rho_g M_{N_2}}{\partial t} + \frac{\partial \rho_g v_x M_{N_2}}{\partial x} + \frac{\partial \rho_g v_y M_{N_2}}{\partial y} + \frac{\partial \rho_g v_z M_{N_2}}{\partial z} = \frac{\partial}{\partial x}\left[\rho_g D \frac{\partial M_{N_2}}{\partial x}\right] + \frac{\partial}{\partial y}\left[\rho_g D \frac{\partial M_{N_2}}{\partial y}\right] + \frac{\partial}{\partial z}\left[\rho_g D \frac{\partial M_{N_2}}{\partial z}\right], \quad (14)$$

$$\frac{\partial \rho_g M_{H_2O}}{\partial t} + \frac{\partial \rho_g v_x M_{H_2O}}{\partial x} + \frac{\partial \rho_g v_y M_{H_2O}}{\partial y} + \frac{\partial \rho_g v_z M_{H_2O}}{\partial z} = \frac{\partial}{\partial x}\left[\rho_g D \frac{\partial M_{H_2O}}{\partial x}\right] + \frac{\partial}{\partial y}\left[\rho_g D \frac{\partial M_{H_2O}}{\partial y}\right] + \frac{\partial}{\partial z}\left[\rho_g D \frac{\partial M_{H_2O}}{\partial z}\right] + \Phi_{H_2O}. \quad (15)$$

The state equation of the gas phase has the form:

$$\rho_g = \frac{p_g}{RT_g}\left(\frac{M_{CO}}{W_{CO}} + \frac{M_{O_2}}{W_{O_2}} + \frac{M_{CO_2}}{W_{CO_2}} + \frac{M_{N_2}}{W_{N_2}} + \frac{M_{H_2O}}{W_{H_2O}}\right)^{-1}. \quad (16)$$

The Navier–Stokes equations were used in the Oberbeck–Boussinesq approximation to describe the flow field, which in the rectangular coordinate system has the form [71,72]:

$$\frac{\partial \rho v_x}{\partial x} + \frac{\partial \rho v_y}{\partial y} + \frac{\partial \rho v_z}{\partial z} = 0, \quad (17)$$

$$\frac{\partial \rho v_x}{\partial t} + \frac{\partial \rho v_x^2}{\partial x} + \frac{\partial \rho v_y v_x}{\partial y} + \frac{\partial \rho v_z v_x}{\partial z} = -\frac{\partial p}{\partial x} + \frac{\partial}{\partial x}\left[\mu \frac{\partial v_x}{\partial x}\right] + \frac{\partial}{\partial y}\left[\mu \frac{\partial v_x}{\partial y}\right] + \frac{\partial}{\partial z}\left[\mu \frac{\partial v_x}{\partial z}\right], \quad (18)$$

$$\frac{\partial \rho v_y}{\partial t} + \frac{\partial \rho v_x v_y}{\partial x} + \frac{\partial \rho v_y^2}{\partial y} + \frac{\partial \rho v_z v_x}{\partial z} = -\frac{\partial p}{\partial y} + \frac{\partial}{\partial x}\left[\mu \frac{\partial v_y}{\partial x}\right] + \frac{\partial}{\partial y}\left[\mu \frac{\partial v_y}{\partial y}\right] + \frac{\partial}{\partial z}\left[\mu \frac{\partial v_y}{\partial z}\right], \quad (19)$$

$$\frac{\partial \rho v_z}{\partial t} + \frac{\partial \rho v_x v_z}{\partial x} + \frac{\partial \rho v_y v_z}{\partial y} + \frac{\partial \rho v_z^2}{\partial z} = -\frac{\partial p}{\partial z} + \frac{\partial}{\partial x}\left[\mu \frac{\partial v_z}{\partial x}\right] + \frac{\partial}{\partial y}\left[\mu \frac{\partial v_z}{\partial y}\right] + \frac{\partial}{\partial z}\left[\mu \frac{\partial v_z}{\partial z}\right] + (\rho - \rho_0)g. \quad (20)$$

It was assumed in the flow structure model that the air movement near the surface occurred in a laminar regime, while the effect of the influx of pyrolysis products and evaporated water into the gas phase on the flow structure was negligibly small.

Boundary conditions for a system of differential equations are determined by the boundary type. Uniform distributions of temperature and mass fraction of the gas phase components are stated for the windward side. Soft boundary conditions were set for the leeward, lateral side, and upper boundaries. The wind speed was assumed to be constant and independent of height. The temperatures of the wood layer and air were equal at

the initial moment of time. The firebrand temperature corresponded to the smoldering temperature of tree bark and twigs of conifers.

The equations in this work presented a completely closed system of equations which, under appropriate boundary and initial conditions and known properties of the medium, determined the main characteristics of heat exchange. The equations were solved numerically using the finite volume method. In accordance with this method, finite difference equations were obtained by integrating differential equations over control volumes containing points of a finite difference grid. The calculations were carried out on a grid with 2000 points in the OX direction, 2500 nodes in the OY direction, and 1500 nodes in the OZ direction. The grid was thickened near the firebrands. The continuity equation was satisfied using the SIMPLEC algorithm. It was considered that the iteration convergence was achieved if the root–mean–square discrepancy for all variables did not exceed 1%. A series of calculations were performed on sequences of refining grids to assess the accuracy of the calculations. The test results showed that a 2-fold decrease in the base grid step along the axial and radial coordinates led to a change in the values of the main variables by no more than 1%.

The ignition of the wood layer by two firebrands of forest combustible material was simulated during the calculations. Pine twigs were chosen as the firebrands' model.

To study the considered process, the evolution of isotherms on the surface of the wood layer was analyzed, along with the change in temperature over time at three characteristic points. Point (A) was chosen at a distance of 3 mm from the windward edge of the firebrand lying upstream. Point (B) was located between the firebrands at an equal distance from their edges. Point (C) was chosen at a distance of 3 mm from the leeward edge of the firebrand lying downstream. All control points were located on the OX axis. The layout of firebrands and control points is shown in Figure 2 (axes were chosen in the wind direction).

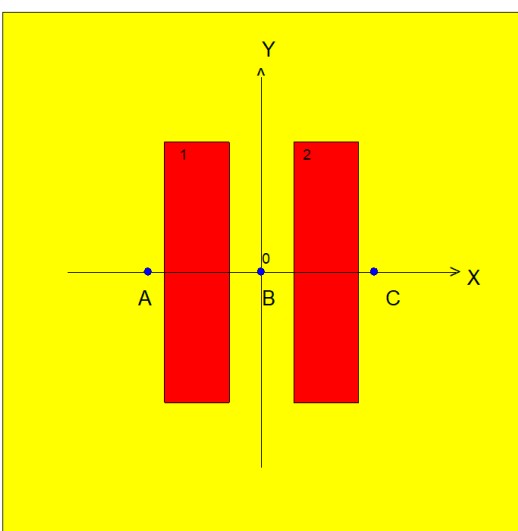

**Figure 2.** Scheme of control points.

## 3. Results and Discussion

The following analysis of the results of mathematical modeling was carried out. Figure 3 shows the temperature change over time at the control point when two thin firebrands hit the surface simultaneously. One can see from Figure 3 that the heat exchange in thin firebrands is carried out in a low-temperature regime. The temperature of the wood layer increases at first, which is associated with the wood being heated by the firebrand. As a result of heat exchange with the environment, cooling then occurs after the thermal energy reserves are exhausted and the temperature decreases. The results of calculations show that the firebrands' length has practically no effect on the process of heating the wood layer during the first three seconds. However, since there is an increase in the length

of the firebrands, an increase in the thermal energy reserves occurs and the transition time for heating the wood layer increases. As can be seen from Figure 3, the coordinate of the maximum temperature curve shifts to the right with a rise in L. At the same time, an increase in the maximum temperature at the control point is observed. To initiate the ignition process, a glowing firebrand must have significant thermal energy, which depends on the size of the firebrands. The calculation results show that the ignition of the wood layer by two thin firebrands with a diameter of d < 5 mm located at a distance of more than 3 mm is impossible. However, in the case of firebrands with a diameter of more than 5 mm, a high-temperature heat exchange regime becomes possible, leading to ignition of the wood surface.

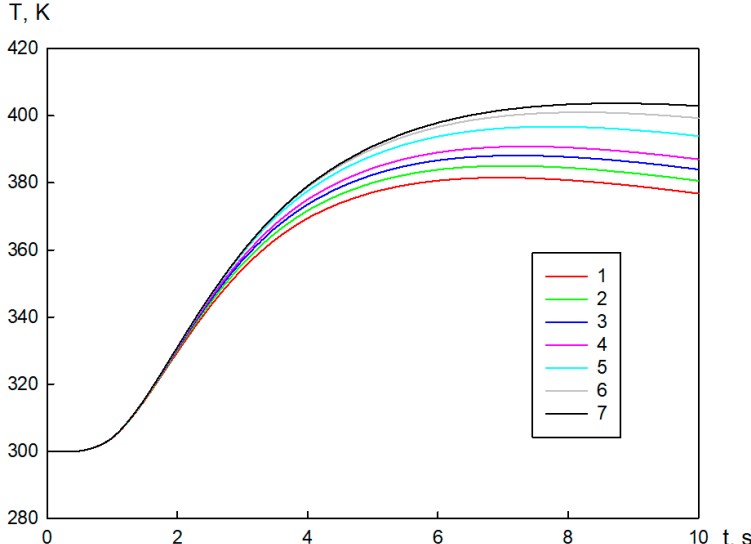

**Figure 3.** Temperature change over time: Δt = 0, d = 4.5 mm, h = 40 mm, 1—L = 140 mm, 2—L = 100 mm, 3—L = 84 mm, 4—L = 76 mm, 5—L = 68 mm, 6—L = 62 mm, 7—L = 58 mm.

The effect of the firebrands' length on the heat exchange with the wood layer and the ignition process is shown in Figure 4. A low-temperature regime is realized for firebrands of small length. As the firebrand length increases, the temperature of the wood layer in its vicinity increases. The wood surface ignites if the firebrand length exceeds a critical value, which depends on other firebrands' sizes and heat transfer conditions.

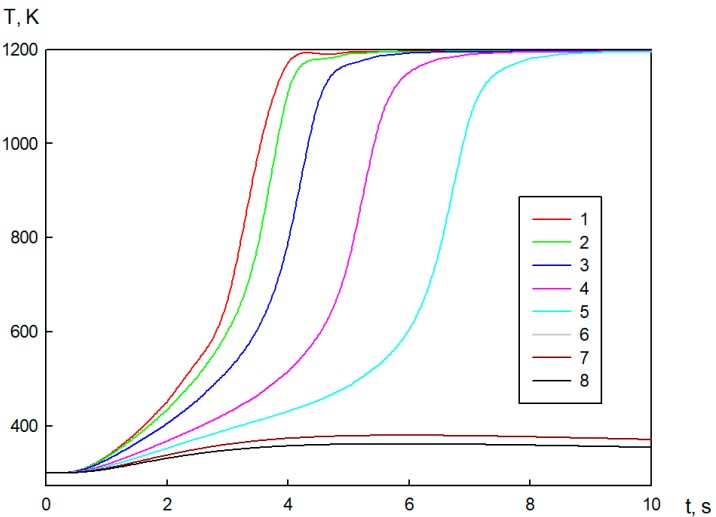

**Figure 4.** Temperature change over time: Δt = 0, d = 15 mm, h = 40 mm, 1—L = 59 mm, 2—L = 49 mm, 3—L = 39 mm, 4—L = 29 mm, 5—L = 26 mm, 6—L = 22 mm, 7—L = 18 mm, 8—L = 14 mm.

A further increase in the firebrand length leads to a decrease in the induction period and earlier ignition. Curves 1–6 characterize the heat exchange in thick firebrands and refer to the high temperature regime. The temperature change over time is close to linear in the initial time period, which corresponds to the heating and drying of the wood layer. An exponential rise in temperature occurs after the wood layer is ignited. The temperature rate increase slows down as the burning increases. The temperature approaches the adiabatic combustion temperature. However, it does not reach its value due to heat exchange. The temperature then decreases at the considered points as the burning proceeds. The cooling process that follows the wood layer burning out occurs at sufficiently long times and is not shown in the Figure.

Figure 5 shows the effect of distances between firebrands on heat exchange and ignition processes. As a single firebrand of the considered sizes does not initiate the ignition of the wood layer, the high-temperature heat exchange regime in the vicinity of the considered point is realized as a result of the energy supply from both firebrands. The calculation results show that two thin twigs located in close proximity to each other ($h = 0$) cause ignition of the wood layer. At the same time, the processes of heating and drying proceed with high intensity, which leads to rapid wood ignition.

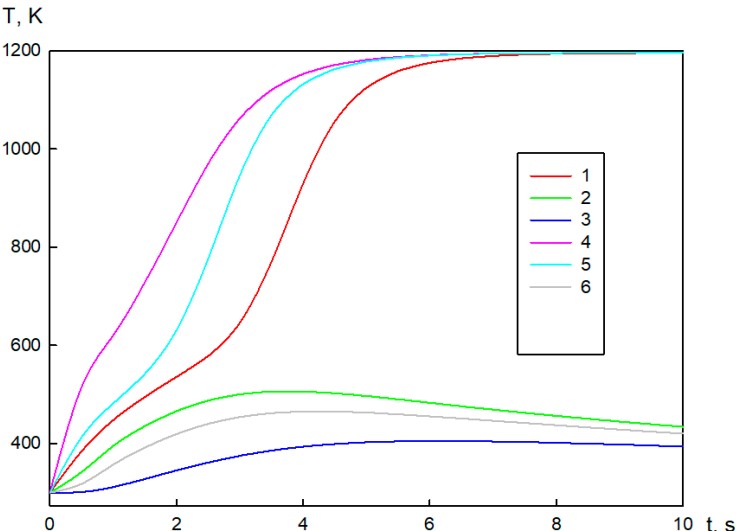

**Figure 5.** Temperature change over time: $\Delta t = 0$, d = 15 mm, L = 22 mm, h = 40 mm, 1—h = 32 mm, 2—h = 45 mm, 3—h = 59 mm, 4—h = 12 mm, 5—h = 26 mm, 6—h = 55 mm.

If there is some distance between the firebrands, part of their thermal energy will be spent on heating the gap between them. As a result, the heating rate of the outer domain slows down. However, the energy supplied from them may be sufficient for ignition if the firebrands are located close to each other. Thus, for closely spaced firebrands, their thermal interaction with the wood layer occurs in the ignition regime. The ignition time increases nonlinearly with increasing distance between them and tends to infinity at $h \to h_*$. The distance between the firebrands $h_*$ determines the boundary of the transition from the high-temperature regime to the low-temperature regime. In the low-temperature regime (at $h \geq h_*$), the wood layer cools down after a slight initial heating.

Figure 6 shows the ignition process of the wood layer with two thick twigs. At the initial moment of time, the wood layers adjacent to it are heated as a result of heat exchange between the wood layer and a twig. The heating area around each twig increases. The interaction processes of smoldering twigs with the heating layer of the wood layer are intensified when the closing of thermal fronts occurs. As a result, the drying process is intensified. The pyrolysis reaction is initiated after the moisture is removed from the wood. The gaseous pyrolysis products mix with atmospheric oxygen and an inflammation occurs. A high-temperature combustion zone is formed near the surface of the firebrand, which

increases in size with time. After the pyrolysis products burn out, the afterburning of the condensed products occurs in the smoldering regime.

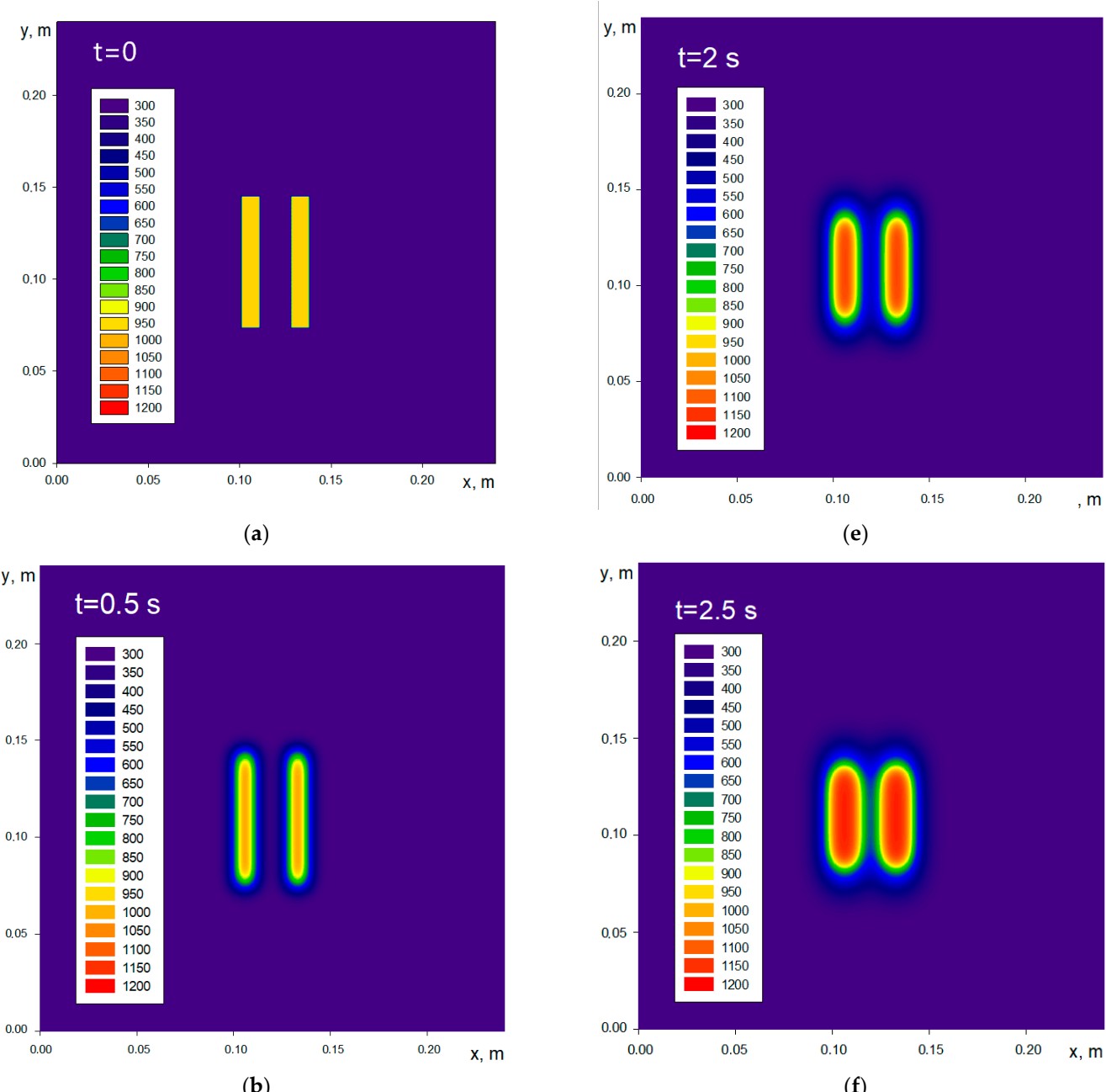

**Figure 6.** *Cont*.

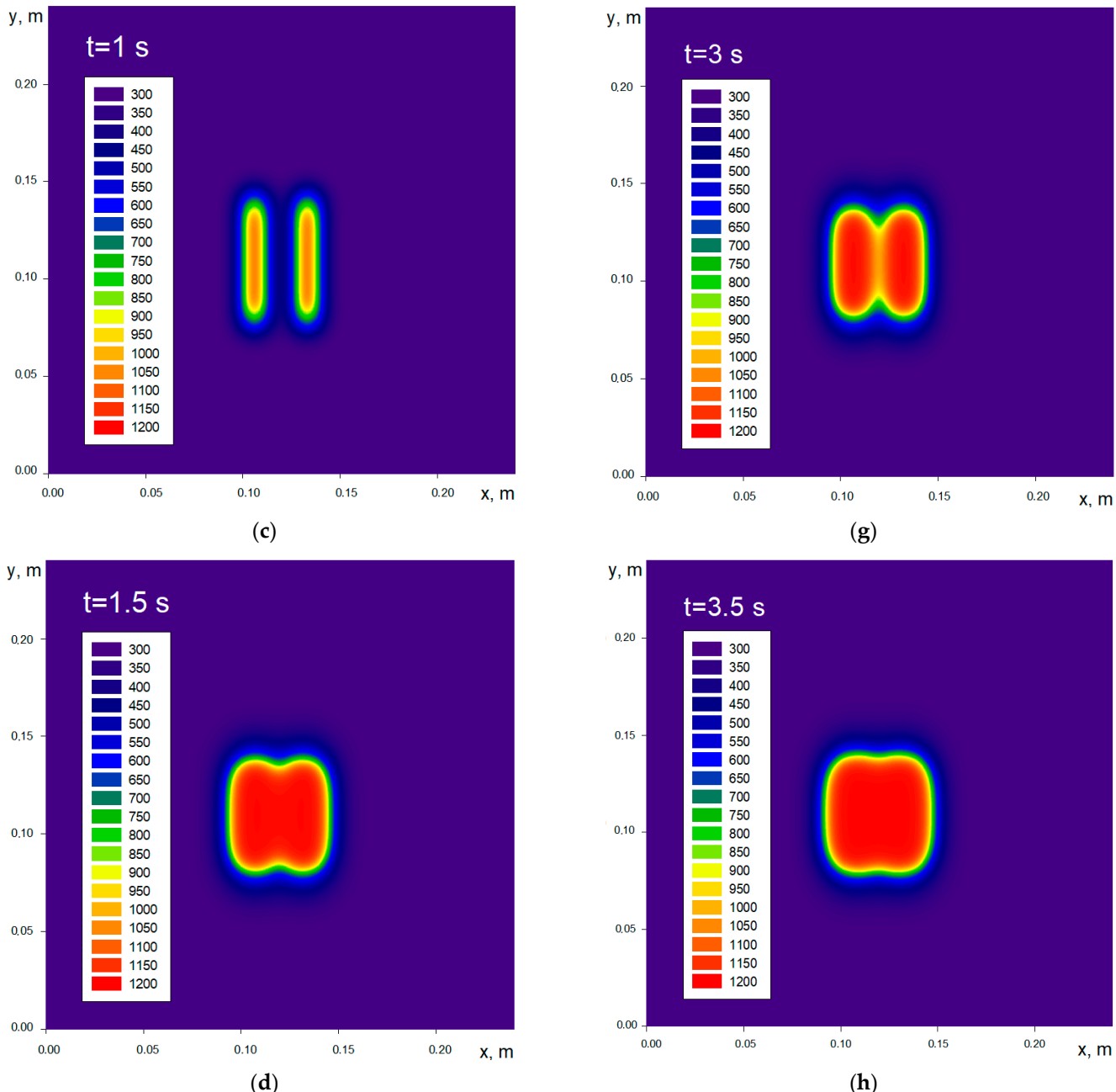

**Figure 6.** Isotherms on the wood layer surface: at different moment L = 69 mm, d = 7.5 mm, h = 16.6 mm, Δt = 0.

The non-simultaneous reaching of the wood surface by igniting firebrands is considered (Figure 7). While carrying out the main series of studies, it was assumed that the wood surface is first reached by firebrand *1*, and later by firebrand *2*, which is located on the leeward side with respect to firebrand *1*.

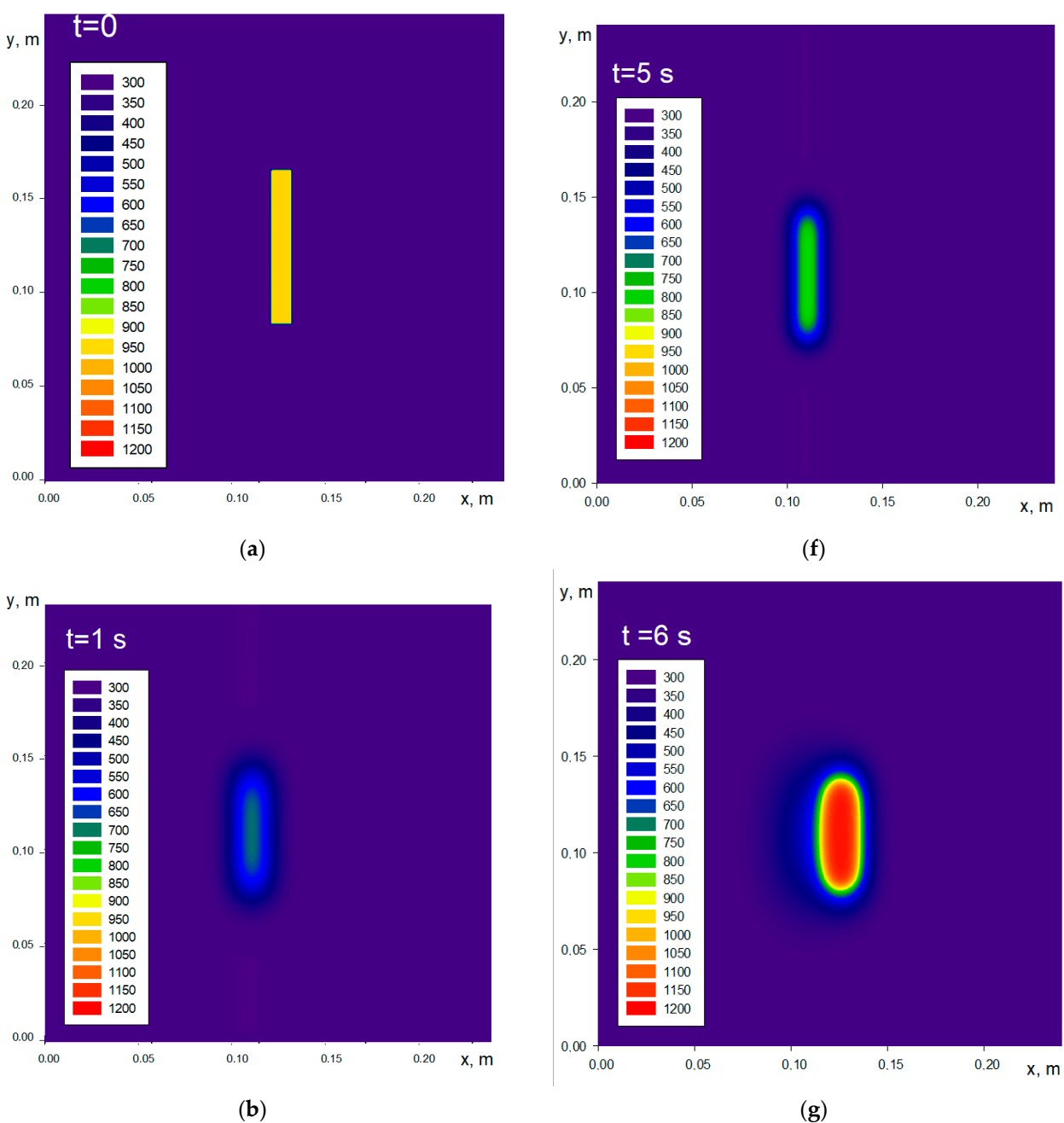

**Figure 7.** *Cont.*

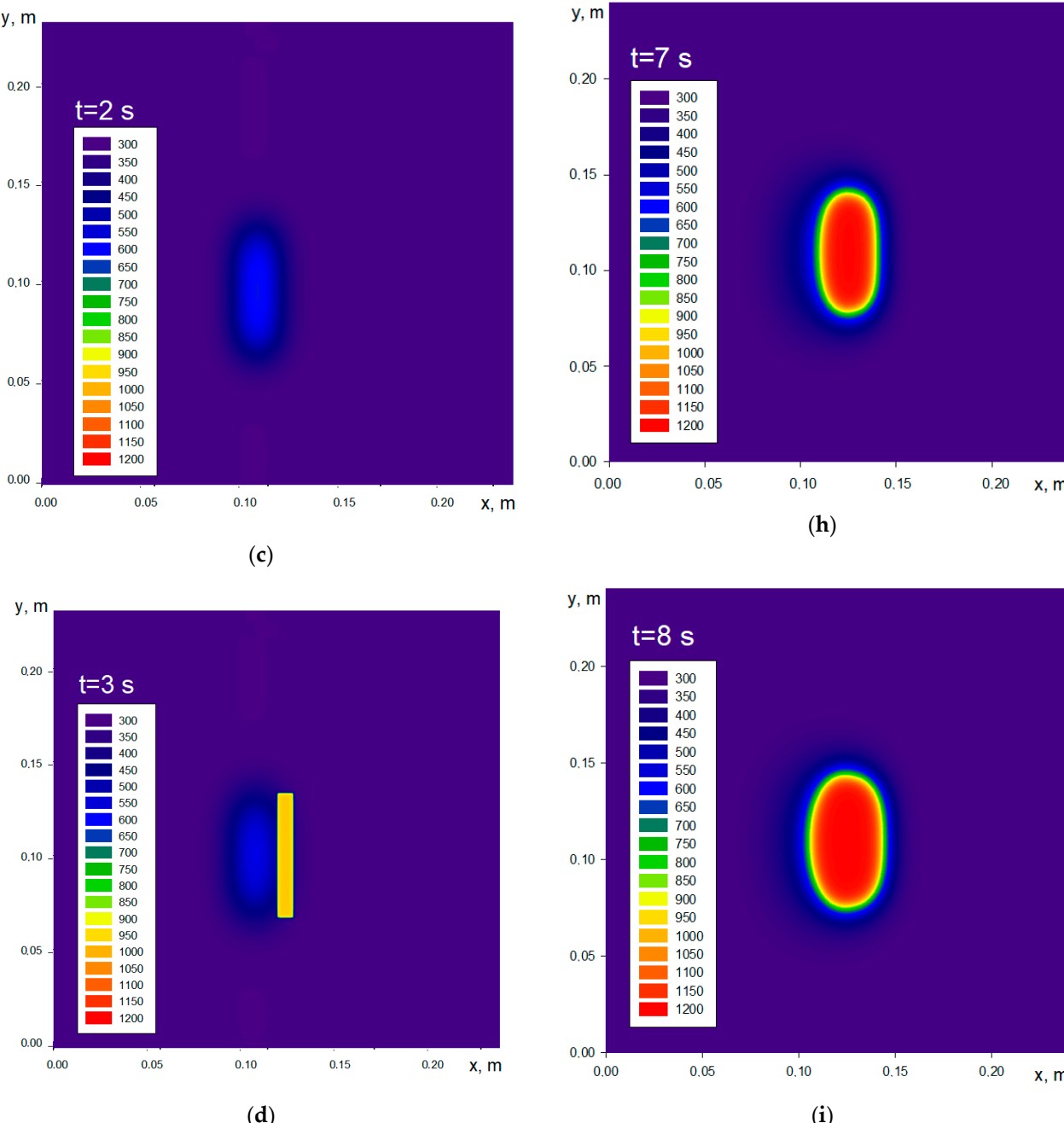

**Figure 7.** *Cont.*

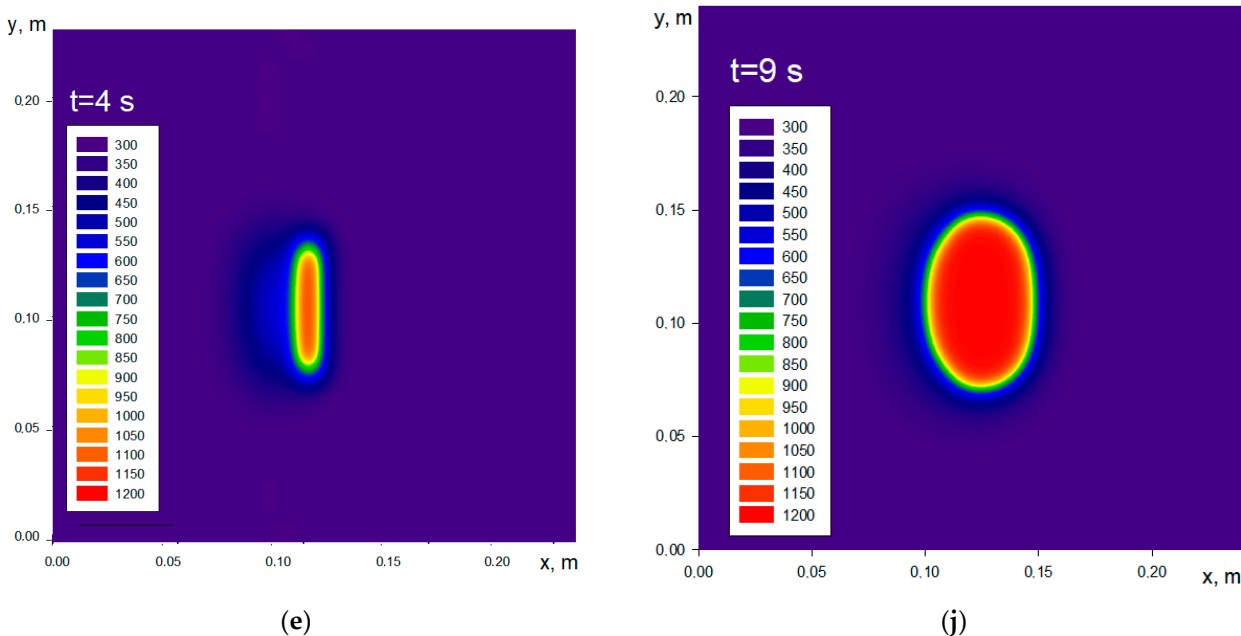

**Figure 7.** Isotherms on the wood layer surface at difference moment: L = 69 mm, d = 6.5 mm, h = 8.5 mm.

The thermal energy stored in a small firebrand is not enough to initiate ignition. Therefore, the interaction of such firebrand with a wood surface occurs in a low-temperature regime. The adjacent wood layers are heated as a result of heat exchange with the firebrand (Figure 7b–d). The heating area increases in the initial period of time. An increase in the area of the heating zone leads to the cooling of the firebrand. As a result, the intensity of smoldering in the firebrand weakens and eventually stops. Subsequently, heat exchange with the environment leads to the cooling of the firebrand and the wood layer adjacent to it. The heat transfer regime changes significantly after the hitting of the surface by the second firebrand. The wood layer is heated and dried by the time the second firebrand hits the surface. The pyrolysis reaction is intensified in the dried and heated wood layer. The gaseous pyrolysis products mix with atmospheric oxygen, the temperature in the channel wall reaches the adiabatic combustion temperature $T_a$, and an inflammation occurs. A high-temperature combustion zone is formed near the firebrand surface, which increases in size over time.

The effect of geometrical parameters of firebrands on heat exchange and ignition is also considered. Figure 8 shows the temperature change over time at three control points: A, B, C; these points are calculated for various firebrands' lengths. One can see from the Figure 8 below that the heat exchange in firebrands of small length is carried out in a low-temperature regime. The temperature of the firebrands approaches the ambient temperature in this case. The heat exchange in long firebrands proceeds in a high-temperature regime, leading to the ignition of the wood layer.

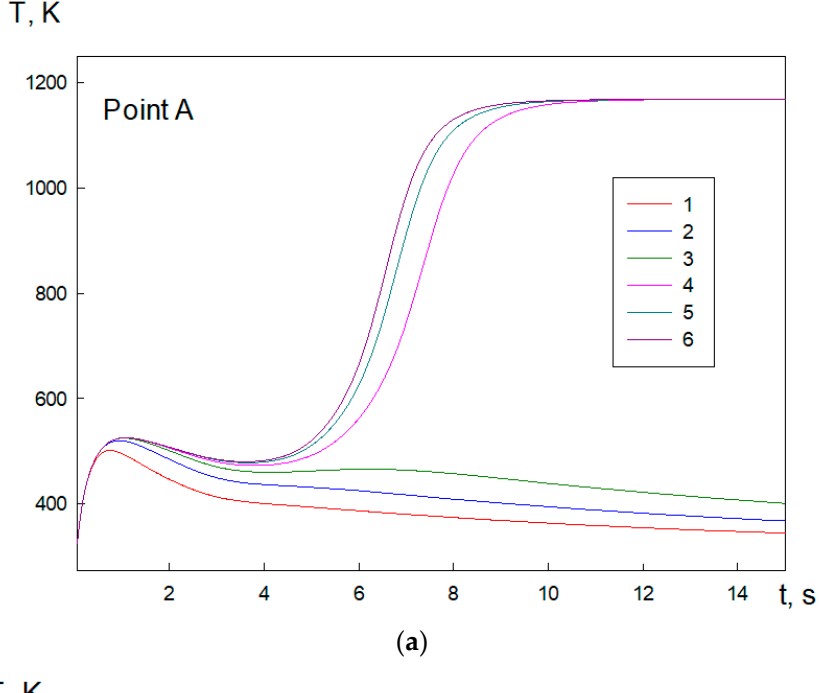

(**a**)

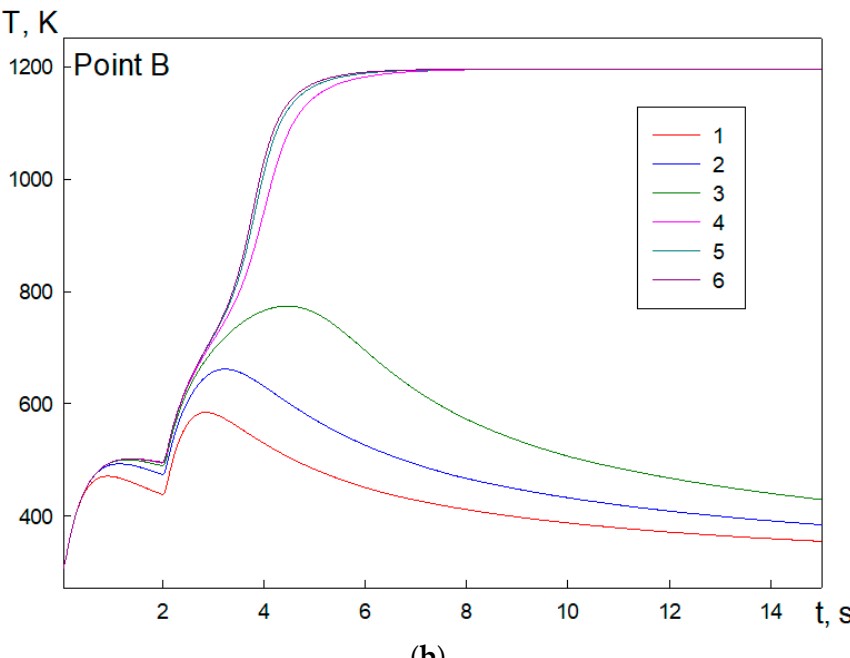

(**b**)

**Figure 8.** *Cont.*

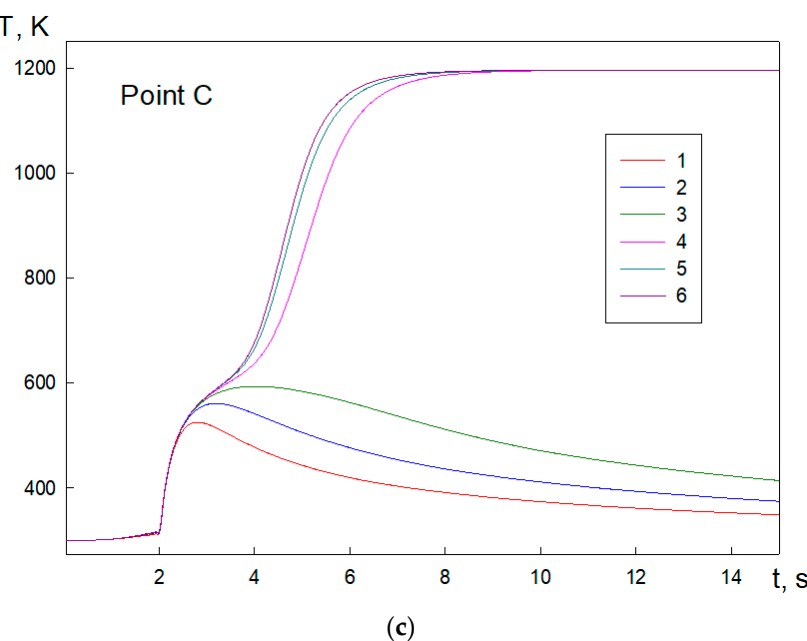

**Figure 8.** Temperature change over time: Δt = 2 s, d = 5.5 mm, h = 6.5 mm, 1 —L = 19 mm, 2 – L = 29, 3– L = 39, 4 – L = 49, 5 – L = 59, 6 – L = 69.

The temperature change at the control points in the initial period of time (until the second firebrand lands on the surface) is typical for the low-temperature regime. The wood layers adjacent to the firebrands are cooled after a slight heating. In this case, the firebrand temperature approaches the ambient temperature. The cooling process after the wood layer burns out occurs over a sufficiently long time period and is not shown in the Figure.

After the firebrand hits the wood layer, a temperature boundary layer is formed, which spreads over the wood surface over time. After the temperature front reaches the control point, its vicinity begins to warm up. The lowest heating temperature is reached at point *C*, which is the furthest point from the firebrand edge; the highest one is at point *A*, which is closest to the firebrand edge. The temperature rises sharply after the second firebrand hits the wood surface at points *B* and *C*. At point *A*, which is the most distant from the second firebrand, the temperature still decreases for some time. The heating of the wood layer begins only after a period of time equal to $\delta_t = \frac{\lambda_W}{C_w \rho_w} X_A^2$ ($X_A^2$ is the distance from point *A* to the windward edge of the second firebrand). After hitting the surface by the second firebrand, intense heating of the area bounded by these firebrands occurs. Thus, the region of the highest wood temperature is localized. In this case, pyrolysis and ignition occur if the intensity of heating of the wood layer by smoldering firebrands exceeds the intensity of heat losses. Firstly, the wood layers adjacent to the left edge of the second firebrand are ignited, followed by the right edge. The mixing of pyrolysis products with atmospheric oxygen leads to an inflammation, accompanied by a sharp, pike-shaped increase in temperature. After the gaseous pyrolysis products in a current domain burn out, the combustion regime is replaced by a smoldering one. In this case, the pyrolysis area extends into the region external to the firebrand. If the firebrand size is sufficiently large, a flame combustion regime occurs in the gas phase after ignition of the wood layer. After the pyrolysis products burn out, the burning of the condensed products occurs in the smoldering regime.

The analysis of the firebrand diameter effect on the ignition conditions of the wood surface will be considered (Figure 9). The dependences presented in the Figure below are similar to those presented above. A low-temperature heat exchange regime is realized for small-diameter firebrands. Comparison of curves 1, 2, illustrating the low-temperature regime of heat exchange, shows that the temperature of the wood layer slightly increases at all control points with an increase in the firebrand diameter. We also note that for point *C*, the increase in temperature with increasing diameter becomes noticeable only after the

second firebrand hits the surface as the effect of firebrand 1 is small due to its distance from point *C*. A significant supply of thermal energy is required to ignite the wood layer, which can be possessed by large firebrands (curves 3–5). The change in temperature at point *B* for firebrands of average diameter (d = 5.5 mm) will be considered in more detail. In this case, before the second firebrand hits the surface, the surface is primarily heated by the first firebrand. The surface is later cooled as a result of heat transfer. After the second firebrand hit the surface, intensive heating and drying of wood occurs in the vicinity of point *B*. The beginning of the pyrolysis process characterizes the inflection point on the temperature curve and outlet on the horizontal asymptote indicates the transition to ignition. The difference of curve 3 at point *C* is the absence of the initial cooling stage, since the thermal wave from the first firebrand still continues to propagate. Moreover, the duration of the drying period, which precedes the start of pyrolysis, increases. This is explained by the fact that the temperature at point *C* at the moment the second firebrand hits the surface is lower than the temperature at point *B*. The cooling time of the wood layer at point *A* is much longer than at point *B*. At the same time, the start of secondary heating at point *A* occurs when the transition to ignition is carried out at point *B*. The time preceding the ignition of the firebrands decreases with a further increase in the firebrand diameter. In the case of oversized firebrands (d = 7.5 mm), when the surroundings of the first firebrand are intensively heated, pyrolysis and ignition of the wood layer is carried out immediately after the second firebrand hits the surface: first at point *B*, then at *A*, and lastly at *C*.

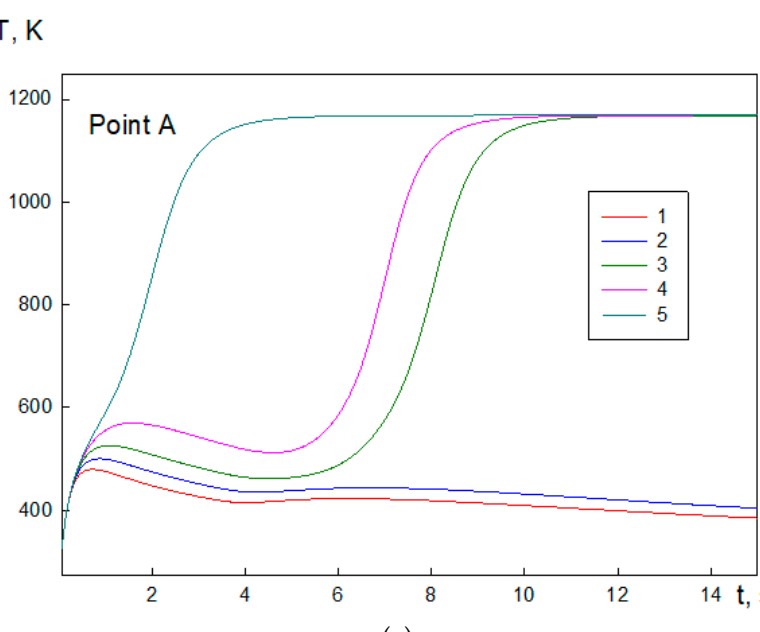

**Figure 9.** *Cont.*

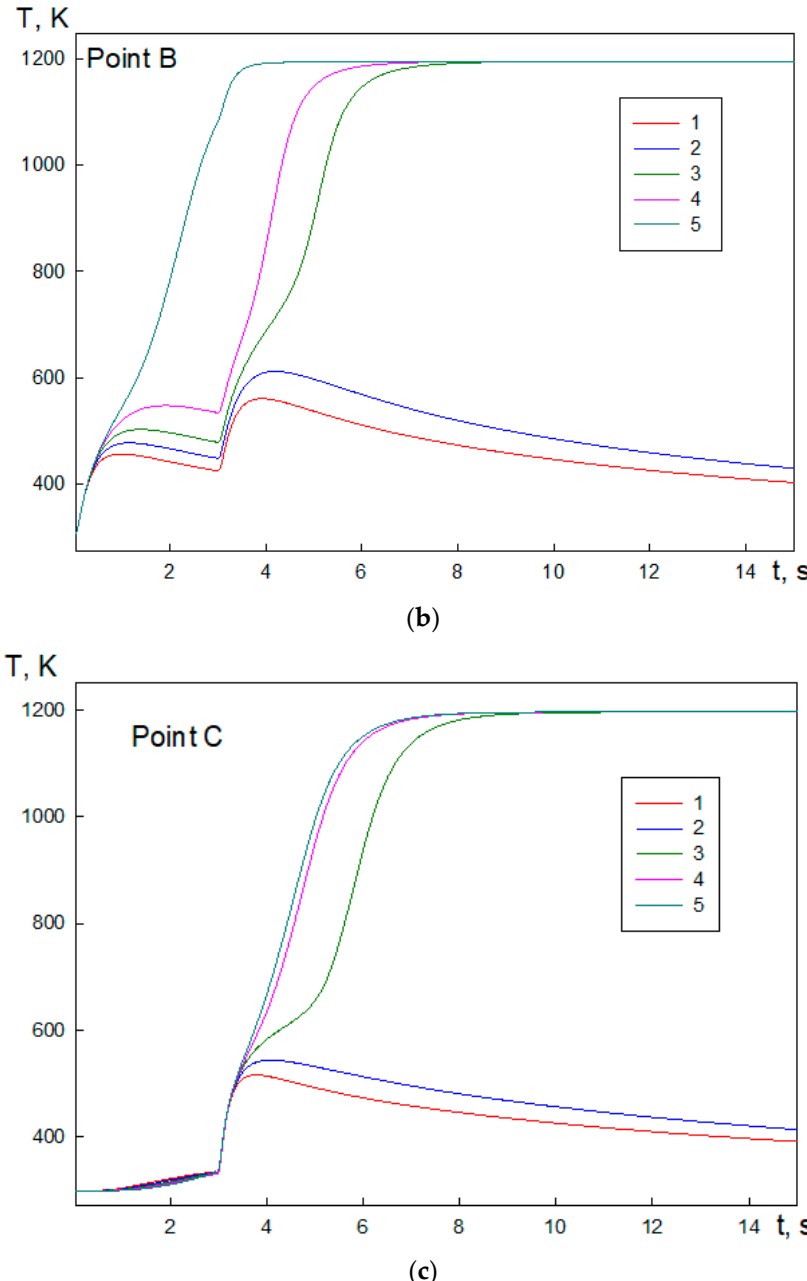

**Figure 9.** Temperature change over time: Δt = 3 s, h = 6.5 mm, 1—L = 69 mm, d = 3.5 mm, 2—d = 4.5 mm, 3—d = 5.5 mm, 4—d = 6.5 mm, 5– d = 7.5 mm.

Figure 10 shows the change in temperature at the control points over time at various distances between the ignition firebrands. One can see from the Figure that the main regularities of the temperature distribution evolution are preserved. At small distances between the firebrands, the process of the second firebrand's interaction with the wood layer after hitting the surface passes successively through all the main stages: heating, drying, pyrolysis, ignition, and combustion. A high-temperature combustion zone is formed near the firebrand surface, which increases in size over time. The heating and drying time increases significantly with an increase in the distance between the firebrands. The period duration of the transition to pyrolysis and ignition at points *B* and, especially, at *C* increases significantly. At point *A*, the cooling time increases significantly. In this case, ignition in the vicinity of this point occurs after the formation of the combustion zone at points *B* and *C*. A low-temperature heat exchange regime is realized at large distances between firebrands. The thermal energy stored in the firebrand is not sufficient for further

heating, drying, and initiation of a chemical reaction. Thermal energy is reduced as a result of heat exchange with air and the wood layer, which leads to cooling and a decrease in temperature. The effect of the distance between the firebrands on the temperature change at the control points becomes insignificant due to their considerable long distance.

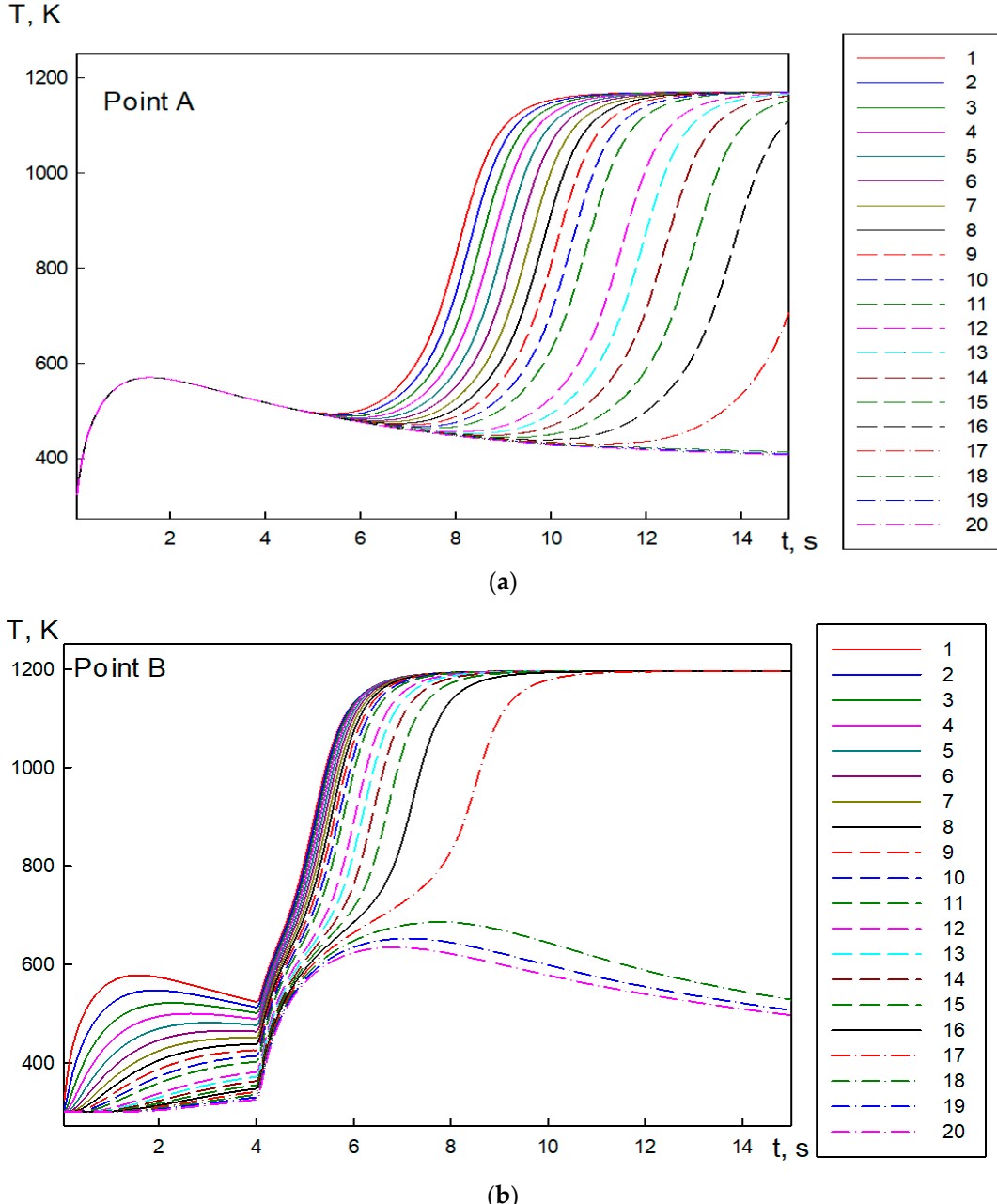

**Figure 10.** *Cont.*

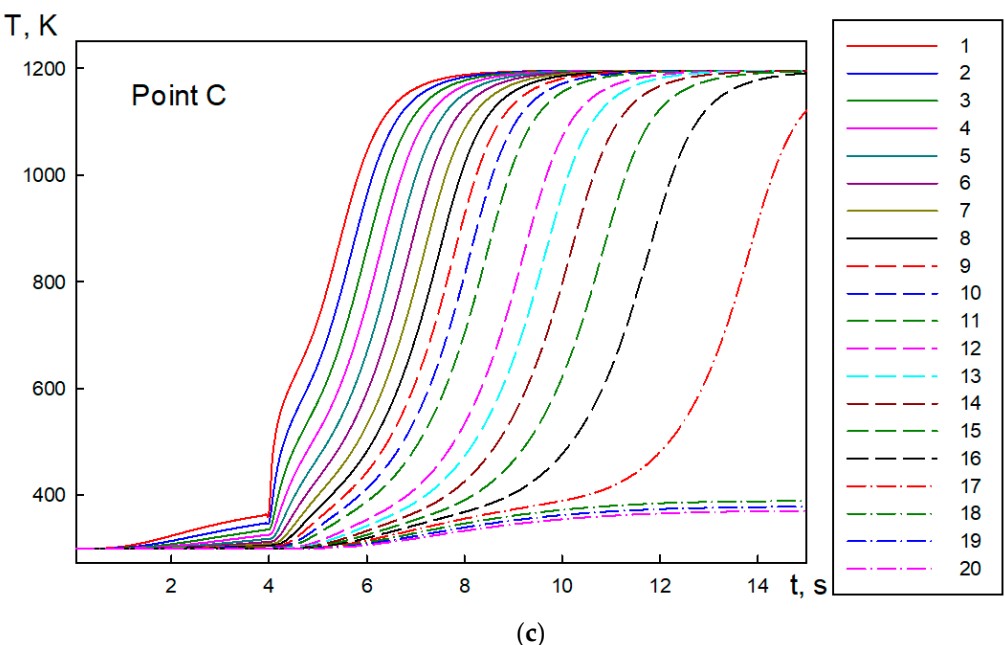

**Figure 10.** Temperature change over time: Δt = 4 s, L= 69 mm, d = 6.5 mm: 1—h = 4.5 mm, 2—h = 6.5 mm, 3—h = 8.5 mm, 4—h = d = 10.5 mm, 5—h = 12.5 mm, 6—h = 14.5 mm, 7—16.5 mm, 8—18.5 mm, 9—h = 20.5 mm, 10—h = 22.5 mm, 11—h = 24.5 mm –, 12—h = 26.5 mm, 13—h = 28.5 mm, 14—h = 30.5 mm, 15—h = 32.5 mm, 16—h = 34.5 mm, 17—36.5 mm, 18—38.5 mm, 19—h=40.5 mm, 20–42.5 mm.

Figure 11 shows the change in temperature at the control points for various times of reaching the surface by the second firebrand Δt.

If the firebrands reach the surface at the same time Δt = 0, the high-temperature heat exchange regime is realized. Curve *1* (points *A* and *C*) clearly shows that there is an increase in temperature in the initial period of time, which indicates the heating of the wood. The drying process then begins and the temperature curves reach saturation, after which pyrolysis and ignition of the wood take place. At point *B*, which is under the effect of two firebrands, these processes proceed much faster, which leads to a decrease in the ignition time. One can see an interesting effect at small delay times Δt < 1 s. If an increase in Δt at point *A* leads only to a slight increase in the drying time, the heating of the point *C* vicinity occurs more intensively as the second firebrand falls on a heated and partially dried surface.

As a consequence of this effect, the ignition time at point *C* decreases compared to the case Δt = 0. The same effect, however, shows itself in a noticeably weaker form at point *B*. A further increase in Δt leads to an increase in the ignition time and a displacement of the temperature curves to the right. This statement is true as long as the high-temperature regime of heat exchange is realized. At long delay times, Δt > 3c firebrand falls on an already cooled surface and cannot cause heating sufficient for ignition. The low-temperature regime is implemented.

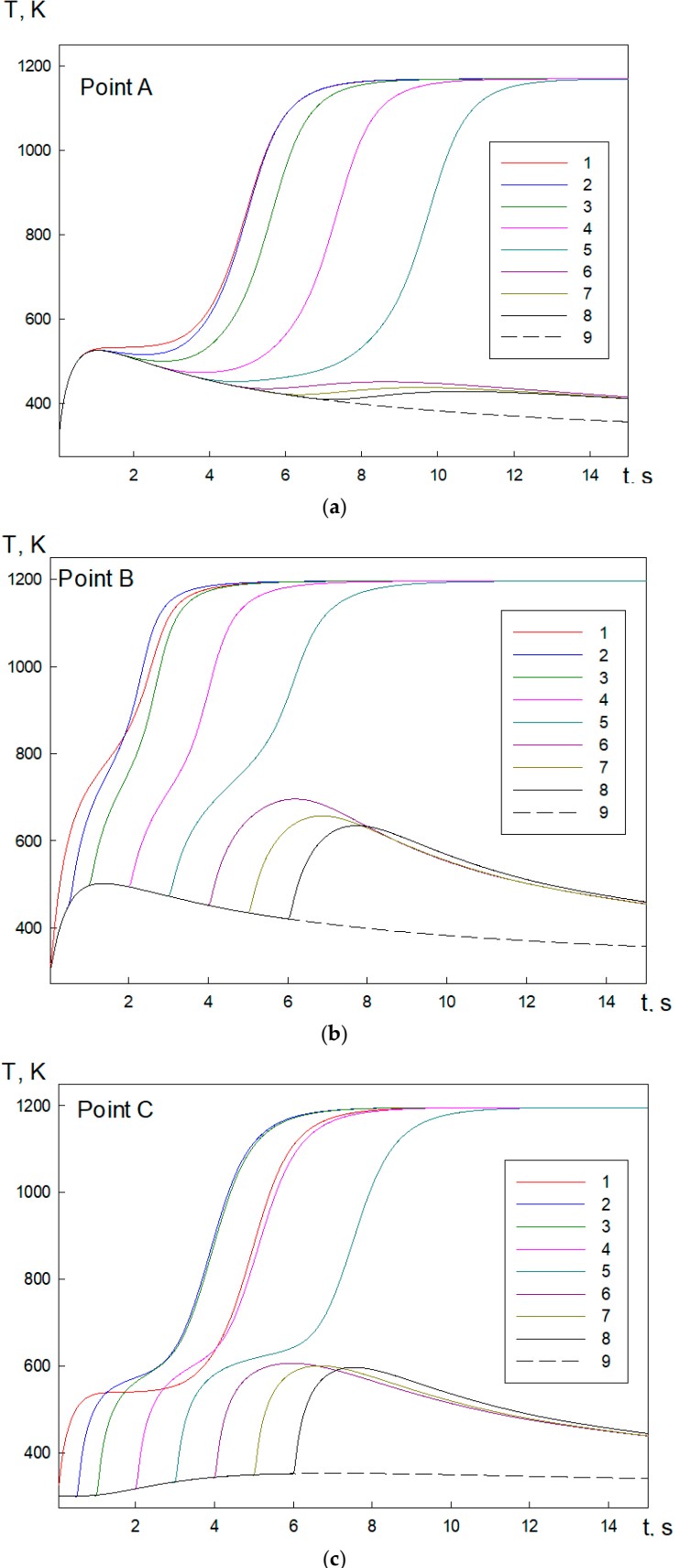

**Figure 11.** Temperature change over time: L = 49 mm, d = 5.5 mm, h = 6.5 mm, 1—Δt = 0, 2—0.5 s, 3—1 s, 4—2 s, 5—3 s, 6—4 s, 7—5 s, 8—6 s, 9—one firebrand.

The results of the carried out studies are summarized in Figure 12, which shows the dependence of the critical delay time Δt on the size of the firebrands and the distance between them. The domain, limited by the coordinate axes and curves in the Figure, corresponds to the implementation of the ignition regime. The outer domain corresponds to the low-temperature regime. One can see from the Figure that the ignition regime becomes possible with an increase in the firebrand size and a longer delay time Δt. The wood ignition is possible with an increase in the distance between the firebrands and a decrease in Δt.

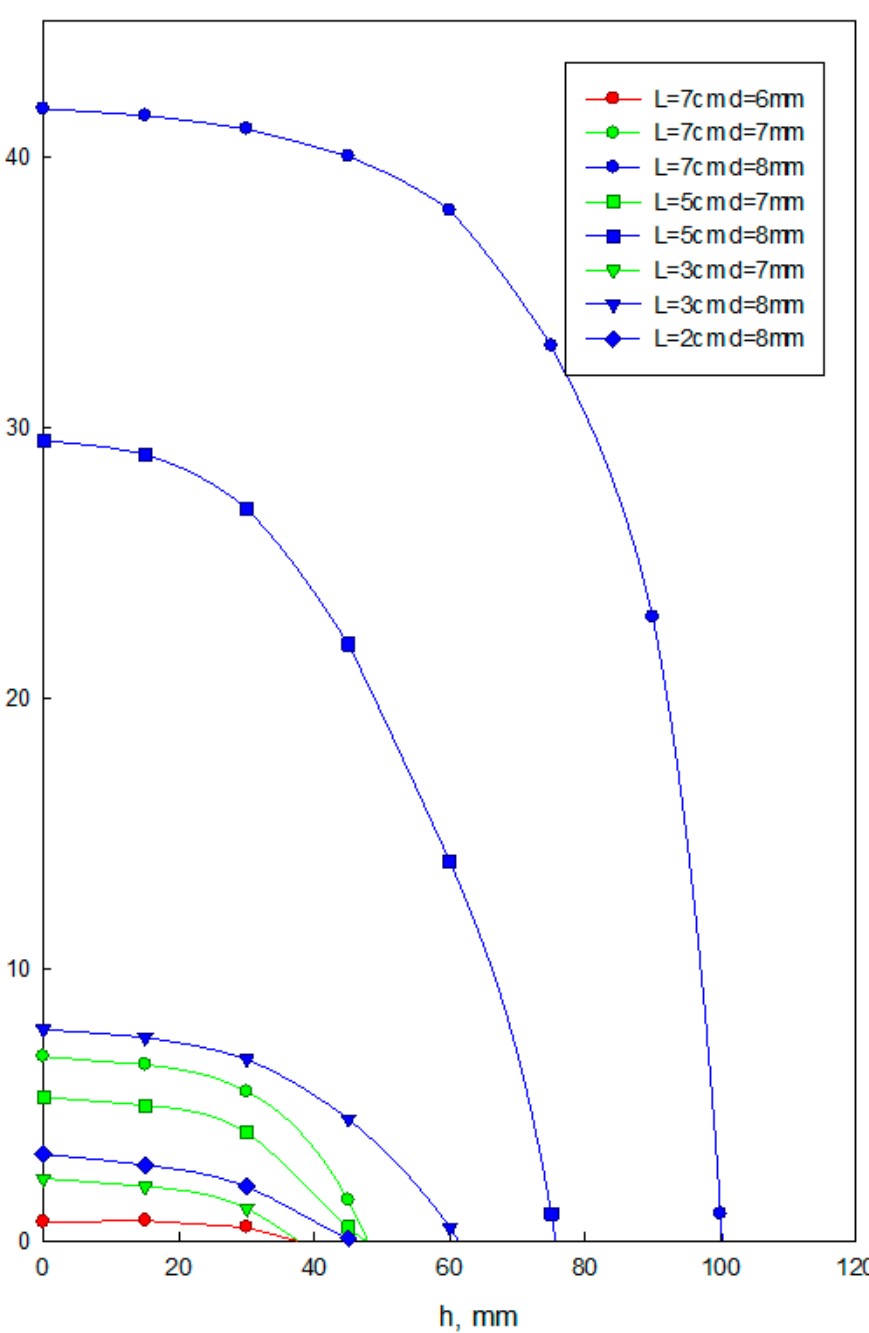

**Figure 12.** Dependence of the critical delay time on the size of the firebrands and the distance between them.

## 4. Conclusions

The model presented considers the heat exchange between the firebrands, wood layer, and the gas phase, as well as moisture evaporation in the firebrands and the diffusion gases of water vapor in the pyrolysis zone. The research results show how the non-simultaneous falling of firebrands onto the wood layer, as well as the distance between them, impacts the ignition and combustion process. The following conclusions can be made as a result of mathematical modeling:

1. The heat stored in firebrands of small sizes is insufficient to initiate the ignition process. The temperature in the wood layer, due to conductive heat exchange, slightly increases at first. Afterwards, it begins to decrease as a result of heat exchange with the surrounding air and the wood layer. Intensive heat exchange with the environment of small size firebrands leads to the end of firebrand smoldering and its cooling. If the firebrand size reaches a critical value, then the pyrolysis process begins in the area adjacent to it.

2. The effect of a set of firebrands (thin twigs) on the process of the wood layer ignition allows one to conclude the following. With a decrease in firebrand size, ignition of wood is possible with a decrease in the distance between the firebrands. With an increase in firebrand size at the same distance between them, the ignition regime becomes possible with a longer delay time $\Delta t$. With a decrease in the distance between the firebrands, the ignition of wood is possible with an increase in $\Delta t$. Results from this study demonstrate the significant influence of spacing between the firebrands on the ignition and the burning behavior of the wood layer. It is note that the narrow spacing between the firebrands significantly increases the likelihood of fire on a building material.

## 5. Limitations

1. The blown air flow was assumed to be laminar. Turbulence was not considered.
2. The wind speed was assumed to be constant over time.
3. The wind direction was chosen to be perpendicular to the long side of the firebrand.
4. The firebrands were assumed to have a regular shape (rectangular parallelepiped) with uniform thermophysical properties. The temperature distribution in the firebrands at the initial moment of time was not taken into account (the temperature at all points of the firebrand had the same value).
5. Ideal contact of the firebrand surface with the wood layer was assumed.

**Author Contributions:** Conceptualization, D.K., O.M. and E.L.; methodology, D.K. and O.M.; validation, D.K., E.L. and O.D.; formal analysis, D.K. and O.M.; investigation, D.K., O.M. and A.L.; writing—original draft preparation, D.K., O.D. and O.M.; writing—review and editing, D.K., E.L. and O.M.; visualization, D.K., O.M., O.D. and A.L.; supervision, O.M.; project administration, D.K.; funding acquisition, D.K. All authors have read and agreed to the published version of the manuscript.

**Funding:** This research was funded by RUSSIAN SCIENCE FOUNDATION, grant number 20-71-10068. Link to information about the project: https://rscf.ru/en/project/20-71-10068/ (accessed on 11 January 2022).

**Conflicts of Interest:** The authors declare no conflict of interest.

## Nomenclature

| Term | Meaning (Units) |
| --- | --- |
| *Nomenclature* | |
| $C$ | heat capacity ($Jkg^{-1}K^{-1}$) |
| $\rho$ | density ($kg\ m^{-3}$) |
| $T$ | temperature ($K$) |
| $\lambda$ | thermal conductivity ($Wm^{-1}k^{-1}$) |
| $U$ | radiation intensity ($W\ m^{-2}$) |

| | |
|---|---|
| $\delta$ | is the characteristic particle size ($m$) |
| $Q$ | reaction heat of pyrolysis, drying and combustion of coke ($Jkg^{-1}$) |
| $\phi$ | the rate of pyrolysis, drying, and combustion (coke) |
| $\varepsilon^*$ | is the degree of blackness of the wood layer |
| $\beta$ | integral radiation absorption coefficient, ($m^{-1}$) |
| $\chi$ | integral radiation attenuation coefficient, ($m^{-1}$) |
| $\sigma$ | is the black body radiation constant |
| $k$ | pre-exponential factor |
| $M_{hum}$ | relative moisture content of wood sample |
| $M_1$ | mass fraction of dry organic matter |
| $M_{pyr}$ | mass fraction of condensed pyrolysis products |
| $M$ | mass fraction |
| $\dot{h}$ | rate of the enthalpy change, ($Jkg^{-1}s^{-1}$) |
| $E$ | activation energy, ($J\ mole^{-1}$) |
| $v$ | wind speed, ($ms^{-1}$) |
| $g$ | gravity factor, ($m\ s^{-1}$) |
| $\Gamma$ | coefficient of diffusion, ($m^2s^{-1}$) |
| $W$ | molecular mass, ($kg\ mole^{-1}$) |
| $p$ | pressure, $Pa$ |
| $\mu$ | viscosity, ($Pa\ s$) |
| $R_g$ | universal gas constant, ($Jmole^{-1}K^{-1}$) |
| $C_{p,g}$ | heat capacity of gas at constant pressure, ($Pa$) |
| $t$ | time, ($s$) |
| $x, y, z$ | coordinates, ($m$) |
| *Subscripts* | |
| $fb$ | firebrand |
| $f$ | solid phase (wood, firebrands) |
| $td$ | thermal degradation |
| $pyr$ | pyrolysis |
| $dr$ | drying |
| $w$ | wood layer |
| $g$ | gas |
| $1$ | dry organic substance |
| $2$ | water in the liquid-drop condition |
| $3$ | condensed pyrolysis products |
| $O_2$ | oxygen |
| $CO$ | carbon monoxide |
| $CO_2$ | carbon dioxide |
| $N_2$ | nitrogen |
| $H_2O$ | water |
| $O$ | ambient |

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
