# Peer review of "Simulation of the Impact of Firebrands on the Process of the Wood Layer Ignition"

_fire, doi:10.3390/fire6040148_

Round 1

Reviewer 1 Report

This paper reported a method for simulating the process of the wood layer ignition by firebrands. The paper is well organized, but in my opinion in current form it is not acceptable for publication. I plead for major revision.

1. No new theory was produced, the innovation of this paper might not be sufficient.

2. How about the grid independence verification?

3. For the Eqs. 3-5, please explain how to obtain these values of each parameter, such as kdr, Tdr?  

4. As it is a simplified model for solid phase ignition, it is better to clarify the limitations of this study considering only numerical results were provided.

5. The captions of Figs. 3-11 must be more detailed. Further, explanations, e.g., “1-L=58 mm, 2-L=62 mm …” should be marked in the figure, but not in the caption. 

Author Response

Dear Reviewer,

Thank you for your valuable comments and suggestions. I have presented below answers to questions: 

(Point 1): No new theory was produced, the innovation of this paper might not be sufficient.

Response 1: The article novelty concludes in the investigation of the ignition process by firebrands falling on the surface non-simultaneously. As a result of the research, we revealed how the ignition process is affected not only by the size of the firebrands and the distance between them, but also by the non-simultaneity of firebrand shitting the surface. The dependence of the critical delay time on the size of the firebrands and the distance between them was obtained. This dependence is important for assessing the ignition conditions of the wood layer surface.

(Point 2): How about the grid independence verification?

Response 2: The equations in this work present a completely closed system of equations, which, under appropriate boundary and initial conditions and known properties of the medium, determines the main characteristics of heat exchange. The equations were solved numerically using the finite volume method. In accordance with this method, finite difference equations were obtained by integrating differential equations over control volumes containing points of a finite difference grid. The calculations were carried out on a grid with 2000 points in the Ox direction, 2500 nodes in the Oy direction, and 1500 nodes in the Oz direction. The grid was thickened near the firebrands. The continuity equation was satisfied using the SIMPLEC algorithm. It was considered that the iterationconvergence was achieved if the root-mean-square discrepancy for all variables did not exceed 1%. A series of calculations were performed on sequences of refining grids to assess the accuracy of the calculations. The test results showed that a 2-fold decrease in the base grid step along the axial and radial coordinates leads to a change in the values of the main variables by no more than 1%.

(Point 3): For the Eqs. 3-5, please explain how to obtain these values of each parameter, such as kdr, Tdr? 

Response 3: The parameter values were taken from the monograph:

  1. Grishin, A. M. Mathematical modeling of forest fires and new methods of fighting them. FA Albini, Ed. 1997.

(Point 4): As it is a simplified model for solid phase ignition, it is better to clarify the limitations of this study considering only numerical results were provided.

Response 4: 1. The blown air flow was assumed to be laminar. Turbulence was not considered.

  1. The wind speed was assumed to be constant over time.
  2. The wind direction was chosen to be perpendicular to the long side of the firebrand.
  3. The firebrands were assumed to have a regular shape (rectangular parallelepiped) with uniform thermophysical properties. The temperature distribution in the firebrands at the initial moment of time was not taken into account (the temperature at all points of the firebrand had the same value).
  4. Ideal contact of the firebrand surface with the wood layer was assumed.

(Point 5): The captions of Figs. 3-11 must be more detailed. Further, explanations, e.g., “1-L=58 mm, 2-L=62 mm …” should be marked in the figure, but not in the caption.

Response 5: Figures 3-11 have been corrected.

Reviewer 2 Report

Dear Authors,

your study has some flaws that need to be removed and fixed before publication. You should deal with these first:

1. there is no scientific aim, it is not specified.

2. Define "wood layer" and "ignition process". These concepts are critically important to this research

3. There is the section named discussion, but no discussion inside.

4. what the conclusions refer to? Is there a hypothesis or clear aim of t the study.

Minor comments:

use the correct way to cite references

revise symbols of units (s, sec)

it is necessary to reorganize the text, L.171-184 are methodical, L. 185-192 suitable for introduction.

For these reasons, your manuscript must be significantly improved (rewritten) before publication

Author Response

Dear Reviewer,

Thank you for your valuable comments and suggestions. I have presented  answers to questions in attached fire. 

Kind Regards, 
Prof. Oleg Matvienko

Round 2

Reviewer 1 Report

This revised paper is much better, I believe it could be published in the present form.

Reviewer 2 Report

Dear Authors,

your manuscript has been sufficiently improved according to the comments. So I recomend to accept it.